# R-FCN: Object Detection via
# Region-based Fully Convolutional Networks

**Jifeng Dai**
Microsoft Research Asia

**Yi Li**[*]
Tsinghua University

**Kaiming He**
Microsoft Research

**Jian Sun**
Microsoft Research

## Abstract

We present region-based, fully convolutional networks for accurate and efficient object detection. In contrast to previous region-based detectors such as Fast/Faster R-CNN [7, 19] that apply a costly per-region subnetwork hundreds of times, our region-based detector is fully convolutional with almost all computation shared on the entire image. To achieve this goal, we propose position-sensitive score maps to address a dilemma between translation-invariance in image classification and translation-variance in object detection. Our method can thus naturally adopt fully convolutional image classifier backbones, such as the latest Residual Networks (ResNets) [10], for object detection. We show competitive results on the PASCAL VOC datasets (*e.g.*, 83.6% mAP on the 2007 set) with the 101-layer ResNet. Meanwhile, our result is achieved at a test-time speed of 170ms per image, 2.5-20× faster than the Faster R-CNN counterpart. Code is made publicly available at: `https://github.com/daijifeng001/r-fcn`.

## 1 Introduction

A prevalent family [9, 7, 19] of deep networks for object detection can be divided into two subnetworks by the Region-of-Interest (RoI) pooling layer [7]: (i) a shared, "*fully convolutional*" subnetwork independent of RoIs, and (ii) an RoI-wise subnetwork that does not share computation. This decomposition [9] was historically resulted from the pioneering classification architectures, such as AlexNet [11] and VGG Nets [24], that consist of two subnetworks by design — a convolutional subnetwork ending with a spatial pooling layer, followed by several fully-connected (*fc*) layers. Thus the (last) spatial pooling layer in image classification networks is naturally turned into the RoI pooling layer in object detection networks [9, 7, 19].

But recent state-of-the-art image classification networks such as Residual Nets (ResNets) [10] and GoogLeNets [25, 27] are by design *fully convolutional*[2]. By analogy, it appears natural to use all convolutional layers to construct the shared, convolutional subnetwork in the object detection architecture, leaving the RoI-wise subnetwork no hidden layer. However, as empirically investigated in this work, this naïve solution turns out to have considerably *inferior detection accuracy* that does not match the network's *superior classification accuracy*. To remedy this issue, in the ResNet paper [10] the RoI pooling layer of the Faster R-CNN detector [19] is *unnaturally* inserted between two sets of convolutional layers — this creates a deeper RoI-wise subnetwork that improves accuracy, at the cost of lower speed due to the unshared per-RoI computation.

We argue that the aforementioned unnatural design is caused by a dilemma of increasing translation *invariance* for image classification *vs.* respecting translation *variance* for object detection. On one hand, the image-level classification task favors translation invariance — shift of an object inside an image should be indiscriminative. Thus, deep (fully) convolutional architectures that are as translation-

---

[*]This work was done when Yi Li was an intern at Microsoft Research.

[2]Only the last layer is fully-connected, which is removed and replaced when fine-tuning for object detection.

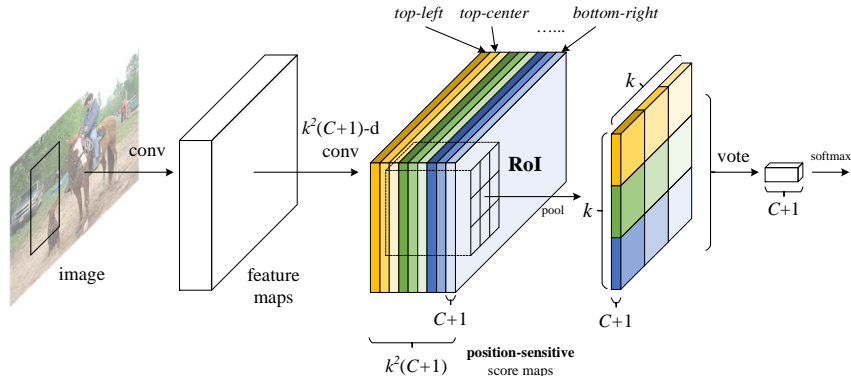

Figure 1: Key idea of **R-FCN** for object detection. In this illustration, there are $k \times k = 3 \times 3$ position-sensitive score maps generated by a fully convolutional network. For each of the $k \times k$ bins in an RoI, pooling is only performed on one of the $k^2$ maps (marked by different colors).

Table 1: Methodologies of *region-based* detectors using **ResNet-101** [10].

|  | R-CNN [8] | Faster R-CNN [20, 10] | R-FCN [ours] |
|---|---|---|---|
| depth of shared convolutional subnetwork | 0 | 91 | 101 |
| depth of RoI-wise subnetwork | 101 | 10 | **0** |

invariant as possible are preferable as evidenced by the leading results on ImageNet classification [10, 25, 27]. On the other hand, the object detection task needs localization representations that are translation-*variant* to an extent. For example, translation of an object inside a candidate box should produce meaningful responses for describing how good the candidate box overlaps the object. We hypothesize that deeper convolutional layers in an image classification network are less sensitive to translation. To address this dilemma, the ResNet paper's detection pipeline [10] inserts the RoI pooling layer into convolutions — this *region-specific* operation breaks down translation invariance, and the post-RoI convolutional layers are no longer translation-invariant when evaluated across different regions. However, this design sacrifices training and testing efficiency since it introduces a considerable number of region-wise layers (Table 1).

In this paper, we develop a framework called *Region-based Fully Convolutional Network* (R-FCN) for object detection. Our network consists of *shared, fully convolutional* architectures as is the case of FCN [16]. To incorporate translation *variance* into FCN, we construct a set of *position-sensitive* score maps by using a bank of specialized convolutional layers as the FCN output. Each of these score maps encodes the position information with respect to a relative spatial position (*e.g.*, "to the left of an object"). On top of this FCN, we append a position-sensitive RoI pooling layer that shepherds information from these score maps, *with no weight (convolutional/fc) layers following*. The entire architecture is learned end-to-end. All learnable layers are convolutional and shared on the entire image, yet encode spatial information required for object detection. Figure 1 illustrates the key idea and Table 1 compares the methodologies among region-based detectors.

Using the 101-layer Residual Net (ResNet-101) [10] as the backbone, our R-FCN yields competitive results of 83.6% mAP on the PASCAL VOC 2007 set and 82.0% the 2012 set. Meanwhile, our results are achieved at a test-time speed of 170ms per image using ResNet-101, which is $2.5\times$ to $20\times$ faster than the Faster R-CNN + ResNet-101 counterpart in [10]. These experiments demonstrate that our method manages to address the dilemma between invariance/variance on translation, and fully convolutional image-level classifiers such as ResNets can be effectively converted to fully convolutional object detectors. Code is made publicly available at: `https://github.com/daijifeng001/r-fcn`.

## 2 Our approach

**Overview.** Following R-CNN [8], we adopt the popular two-stage object detection strategy [8, 9, 6, 7, 19, 1, 23] that consists of: (i) region proposal, and (ii) region classification. Although methods that do not rely on region proposal do exist (*e.g.*, [18, 15]), *region-based* systems still possess leading accuracy on several benchmarks [5, 14, 21]. We extract candidate regions by the Region Proposal

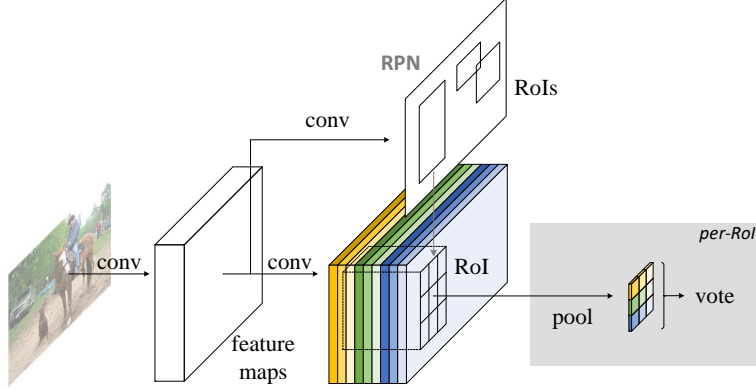

Figure 2: Overall architecture of R-FCN. A Region Proposal Network (RPN) [19] proposes candidate RoIs, which are then applied on the score maps. All learnable weight layers are convolutional and are computed on the entire image; the per-RoI computational cost is negligible.

Network (RPN) [19], which is a fully convolutional architecture in itself. Following [19], we share the features between RPN and R-FCN. Figure 2 shows an overview of the system.

Given the proposal regions (RoIs), the R-FCN architecture is designed to classify the RoIs into object categories and background. In R-FCN, all learnable weight layers are convolutional and are computed on the entire image. The last convolutional layer produces a bank of $k^2$ *position-sensitive score maps* for each category, and thus has a $k^2(C+1)$-channel output layer with $C$ object categories ($+1$ for background). The bank of $k^2$ score maps correspond to a $k \times k$ spatial grid describing relative positions. For example, with $k \times k = 3 \times 3$, the 9 score maps encode the cases of *{top-left, top-center, top-right, ..., bottom-right}* of an object category.

R-FCN ends with a position-sensitive RoI pooling layer. This layer aggregates the outputs of the last convolutional layer and generates scores for each RoI. Unlike [9, 7], our position-sensitive RoI layer conducts *selective* pooling, and each of the $k \times k$ bin aggregates responses from only *one* score map out of the bank of $k \times k$ score maps. With end-to-end training, this RoI layer shepherds the last convolutional layer to learn specialized position-sensitive score maps. Figure 1 illustrates this idea. Figure 3 and 4 visualize an example. The details are introduced as follows.

**Backbone architecture.** The incarnation of R-FCN in this paper is based on ResNet-101 [10], though other networks [11, 24] are applicable. ResNet-101 has 100 convolutional layers followed by global average pooling and a 1000-class *fc* layer. We remove the average pooling layer and the *fc* layer and only use the convolutional layers to compute feature maps. We use the ResNet-101 released by the authors of [10], pre-trained on ImageNet [21]. The last convolutional block in ResNet-101 is 2048-d, and we attach a randomly initialized 1024-d $1\times1$ convolutional layer for reducing dimension (to be precise, this increases the depth in Table 1 by 1). Then we apply the $k^2(C+1)$-channel convolutional layer to generate score maps, as introduced next.

**Position-sensitive score maps & Position-sensitive RoI pooling.** To explicitly encode position information into each RoI, we divide each RoI rectangle into $k \times k$ bins by a regular grid. For an RoI rectangle of a size $w \times h$, a bin is of a size $\approx \frac{w}{k} \times \frac{h}{k}$ [9, 7]. In our method, the last convolutional layer is constructed to produce $k^2$ score maps for each category. Inside the $(i, j)$-th bin ($0 \le i, j \le k-1$), we define a position-sensitive RoI pooling operation that pools only over the $(i, j)$-th score map:

$$r_c(i, j \mid \Theta) = \sum_{(x,y) \in \text{bin}(i,j)} z_{i,j,c}(x + x_0, y + y_0 \mid \Theta)/n. \quad (1)$$

Here $r_c(i, j)$ is the pooled response in the $(i, j)$-th bin for the $c$-th category, $z_{i,j,c}$ is one score map out of the $k^2(C+1)$ score maps, $(x_0, y_0)$ denotes the top-left corner of an RoI, $n$ is the number of pixels in the bin, and $\Theta$ denotes all learnable parameters of the network. The $(i, j)$-th bin spans $\lfloor i\frac{w}{k} \rfloor \le x < \lceil (i+1)\frac{w}{k} \rceil$ and $\lfloor j\frac{h}{k} \rfloor \le y < \lceil (j+1)\frac{h}{k} \rceil$. The operation of Eqn.(1) is illustrated in Figure 1, where a color represents a pair of $(i, j)$. Eqn.(1) performs average pooling (as we use throughout this paper), but max pooling can be conducted as well.

The $k^2$ position-sensitive scores then vote on the RoI. In this paper we simply vote by averaging the scores, producing a $(C + 1)$-dimensional vector for each RoI: $r_c(\Theta) = \sum_{i,j} r_c(i, j \mid \Theta)$. Then we compute the softmax responses across categories: $s_c(\Theta) = e^{r_c(\Theta)} / \sum_{c'=0}^{C} e^{r_{c'}(\Theta)}$. They are used for evaluating the cross-entropy loss during training and for ranking the RoIs during inference.

We further address bounding box regression [8, 7] in a similar way. Aside from the above $k^2(C+1)$-d convolutional layer, we append a sibling $4k^2$-d convolutional layer for bounding box regression. The position-sensitive RoI pooling is performed on this bank of $4k^2$ maps, producing a $4k^2$-d vector for each RoI. Then it is aggregated into a 4-d vector by average voting. This 4-d vector parameterizes a bounding box as $t = (t_x, t_y, t_w, t_h)$ following the parameterization in [7]. We note that we perform class-agnostic bounding box regression for simplicity, but the class-specific counterpart (*i.e.*, with a $4k^2C$-d output layer) is applicable.

The concept of position-sensitive score maps is partially inspired by [3] that develops FCNs for instance-level semantic segmentation. We further introduce the position-sensitive RoI pooling layer that shepherds learning of the score maps for object detection. There is no learnable layer after the RoI layer, enabling nearly *cost-free* region-wise computation and speeding up both training and inference.

**Training.** With pre-computed region proposals, it is easy to end-to-end train the R-FCN architecture. Following [7], our loss function defined on each RoI is the summation of the cross-entropy loss and the box regression loss: $L(s, t_{x,y,w,h}) = L_{cls}(s_{c^*}) + \lambda[c^* > 0]L_{reg}(t, t^*)$. Here $c^*$ is the RoI's ground-truth label ($c^* = 0$ means background). $L_{cls}(s_{c^*}) = -\log(s_{c^*})$ is the cross-entropy loss for classification, $L_{reg}$ is the bounding box regression loss as defined in [7], and $t^*$ represents the ground truth box. $[c^* > 0]$ is an indicator which equals to 1 if the argument is true and 0 otherwise. We set the balance weight $\lambda = 1$ as in [7]. We define positive examples as the RoIs that have intersection-over-union (IoU) overlap with a ground-truth box of at least 0.5, and negative otherwise.

It is easy for our method to adopt online hard example mining (OHEM) [23] during training. Our negligible per-RoI computation enables nearly *cost-free* example mining. Assuming $N$ proposals per image, in the forward pass, we evaluate the loss of all $N$ proposals. Then we sort all RoIs (positive and negative) by loss and select $B$ RoIs that have the highest loss. Backpropagation [12] is performed based on the selected examples. Because our per-RoI computation is negligible, the forward time is nearly not affected by $N$, in contrast to OHEM Fast R-CNN in [23] that may double training time. We provide comprehensive timing statistics in Table 3 in the next section.

We use a weight decay of 0.0005 and a momentum of 0.9. By default we use single-scale training: images are resized such that the scale (shorter side of image) is 600 pixels [7, 19]. Each GPU holds 1 image and selects $B = 128$ RoIs for backprop. We train the model with 8 GPUs (so the effective mini-batch size is $8\times$). We fine-tune R-FCN using a learning rate of 0.001 for 20k mini-batches and 0.0001 for 10k mini-batches on VOC. To have R-FCN share features with RPN (Figure 2), we adopt the 4-step alternating training[3] in [19], alternating between training RPN and training R-FCN.

**Inference.** As illustrated in Figure 2, the feature maps shared between RPN and R-FCN are computed (on an image with a single scale of 600). Then the RPN part proposes RoIs, on which the R-FCN part evaluates category-wise scores and regresses bounding boxes. During inference we evaluate 300 RoIs as in [19] for fair comparisons. The results are post-processed by non-maximum suppression (NMS) using a threshold of 0.3 IoU [8], as standard practice.

**À *trous* and stride.** Our fully convolutional architecture enjoys the benefits of the network modifications that are widely used by FCNs for semantic segmentation [16, 2]. Particularly, we reduce ResNet-101's effective stride from 32 pixels to 16 pixels, increasing the score map resolution. All layers before and on the conv4 stage [10] (stride=16) are unchanged; the stride=2 operations in the first conv5 block is modified to have stride=1, and all convolutional filters on the conv5 stage are modified by the "hole algorithm" [16, 2] ("*Algorithme à trous*" [17]) to compensate for the reduced stride. For fair comparisons, the RPN is computed on top of the conv4 stage (that are shared with R-FCN), as is the case in [10] with Faster R-CNN, so the RPN is not affected by the *à trous* trick. The following table shows the ablation results of R-FCN ($k \times k = 7 \times 7$, no hard example mining). The *à trous* trick improves mAP by 2.6 points.

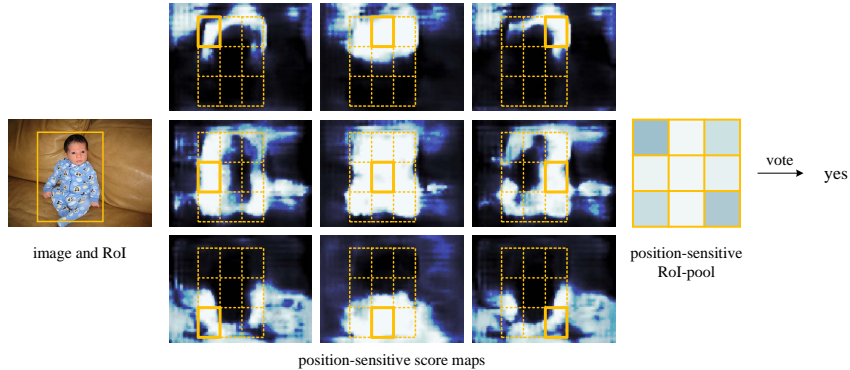

image and RoI

position-sensitive score maps

position-sensitive RoI-pool

vote → yes

Figure 3: Visualization of R-FCN ($k \times k = 3 \times 3$) for the *person* category.

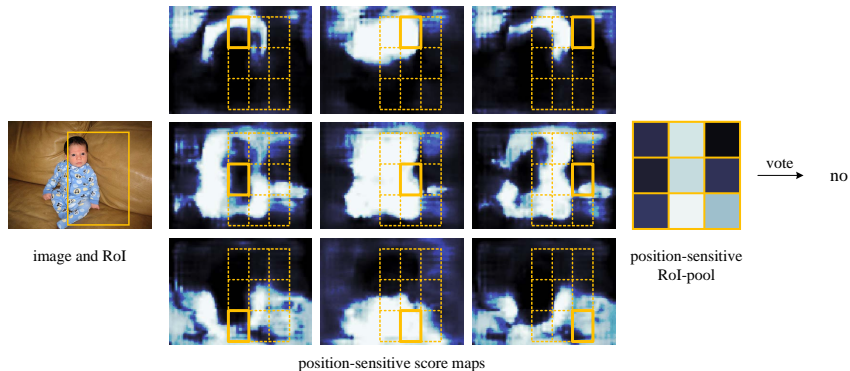

image and RoI

position-sensitive score maps

position-sensitive RoI-pool

vote → no

Figure 4: Visualization when an RoI does not correctly overlap the object.

| R-FCN with ResNet-101 on: | conv4, stride=16 | conv5, stride=32 | conv5, *à trous*, stride=16 |
|---|---|---|---|
| mAP (%) on VOC 07 test | 72.5 | 74.0 | 76.6 |

**Visualization.** In Figure 3 and 4 we visualize the position-sensitive score maps learned by R-FCN when $k \times k = 3 \times 3$. These specialized maps are expected to be strongly activated at a specific *relative position* of an object. For example, the "*top-center-sensitive*" score map exhibits high scores roughly near the top-center position of an object. If a candidate box precisely overlaps with a true object (Figure 3), most of the $k^2$ bins in the RoI are strongly activated, and their voting leads to a high score. On the contrary, if a candidate box does not correctly overlaps with a true object (Figure 4), some of the $k^2$ bins in the RoI are not activated, and the voting score is low.

## 3   Related Work

R-CNN [8] has demonstrated the effectiveness of using region proposals [28, 29] with deep networks. R-CNN evaluates convolutional networks on cropped and warped regions, and computation is not shared among regions (Table 1). SPPnet [9], Fast R-CNN [7], and Faster R-CNN [19] are "*semi-convolutional*", in which a convolutional subnetwork performs shared computation on the entire image and another subnetwork evaluates individual regions.

There have been object detectors that can be thought of as "*fully convolutional*" models. OverFeat [22] detects objects by sliding multi-scale windows on the shared convolutional feature maps; similarly, in Fast R-CNN [7] and [13], sliding windows that replace region proposals are investigated. In these cases, one can recast a sliding window of a single scale as a single convolutional layer. The RPN component in Faster R-CNN [19] is a fully convolutional detector that predicts bounding boxes with respect to reference boxes (anchors) of multiple sizes. The original RPN is class-agnostic in [19], but its class-specific counterpart is applicable (see also [15]) as we evaluate in the following.

Table 2: Comparisons among fully convolutional (or "almost" fully convolutional) strategies using **ResNet-101**. *All competitors in this table use the à trous trick.* Hard example mining is not conducted.

| method | RoI output size ($k \times k$) | mAP on VOC 07 (%) |
|---|---|---|
| naïve Faster R-CNN | $1 \times 1$ | 61.7 |
| | $7 \times 7$ | 68.9 |
| class-specific RPN | - | 67.6 |
| R-FCN (w/o position-sensitivity) | $1 \times 1$ | *fail* |
| R-FCN | $3 \times 3$ | 75.5 |
| | $7 \times 7$ | **76.6** |

Another family of object detectors resort to fully-connected (*fc*) layers for generating holistic object detection results on an entire image, such as [26, 4, 18].

## 4 Experiments

### 4.1 Experiments on PASCAL VOC

We perform experiments on PASCAL VOC [5] that has 20 object categories. We train the models on the union set of VOC 2007 *trainval* and VOC 2012 *trainval* ("07+12") following [7], and evaluate on VOC 2007 *test* set. Object detection accuracy is measured by mean Average Precision (mAP).

**Comparisons with Other Fully Convolutional Strategies**

Though fully convolutional detectors are available, experiments show that it is nontrivial for them to achieve good accuracy. We investigate the following fully convolutional strategies (or "almost" fully convolutional strategies that have only one classifier *fc* layer per RoI), using ResNet-101:

***Naïve Faster R-CNN.*** As discussed in the introduction, one may use *all* convolutional layers in ResNet-101 to compute the shared feature maps, and adopt RoI pooling after the last convolutional layer (after conv5). An inexpensive 21-class *fc* layer is evaluated on each RoI (so this variant is "almost" fully convolutional). The *à trous* trick is used for fair comparisons.

***Class-specific RPN.*** This RPN is trained following [19], except that the 2-class (object or not) convolutional classifier layer is replaced with a 21-class convolutional classifier layer. For fair comparisons, for this class-specific RPN we use ResNet-101's conv5 layers with the *à trous* trick.

***R-FCN without position-sensitivity.*** By setting $k = 1$ we remove the position-sensitivity of the R-FCN. This is equivalent to global pooling within each RoI.

*Analysis.* Table 2 shows the results. We note that the *standard* (not naïve) Faster R-CNN in the ResNet paper [10] achieves 76.4% mAP with ResNet-101 (see also Table 3), which inserts the RoI pooling layer between conv4 and conv5 [10]. As a comparison, the *naïve* Faster R-CNN (that applies RoI pooling *after* conv5) has a drastically lower mAP of 68.9% (Table 2). This comparison empirically justifies the importance of respecting spatial information by inserting RoI pooling between layers for the Faster R-CNN system. Similar observations are reported in [20].

The class-specific RPN has an mAP of 67.6% (Table 2), about 9 points lower than the standard Faster R-CNN's 76.4%. This comparison is in line with the observations in [7, 13] — in fact, the class-specific RPN is similar to a special form of Fast R-CNN [7] that uses dense sliding windows as proposals, which shows inferior results as reported in [7, 13].

On the other hand, our R-FCN system has significantly better accuracy (Table 2). Its mAP (76.6%) is on par with the standard Faster R-CNN's (76.4%, Table 3). These results indicate that our position-sensitive strategy manages to encode useful spatial information for locating objects, without using any learnable layer after RoI pooling.

The importance of position-sensitivity is further demonstrated by setting $k = 1$, for which R-FCN is unable to converge. In this degraded case, no spatial information can be explicitly captured within an RoI. Moreover, we report that naïve Faster R-CNN is able to converge if its RoI pooling output resolution is $1 \times 1$, but the mAP further drops by a large margin to 61.7% (Table 2).

Table 3: Comparisons between Faster R-CNN and R-FCN using ResNet-101. Timing is evaluated on a single Nvidia K40 GPU. With OHEM, $N$ RoIs per image are computed in the forward pass, and 128 samples are selected for backpropagation. 300 RoIs are used for testing following [19].

| | depth of per-RoI subnetwork | training w/ OHEM? | train time (sec/img) | test time (sec/img) | mAP (%) on VOC07 |
|---|---|---|---|---|---|
| Faster R-CNN | 10 | | 1.2 | 0.42 | 76.4 |
| **R-FCN** | 0 | | 0.45 | 0.17 | 76.6 |
| Faster R-CNN | 10 | ✓ (300 RoIs) | 1.5 | 0.42 | 79.3 |
| **R-FCN** | 0 | ✓ (300 RoIs) | 0.45 | 0.17 | **79.5** |
| Faster R-CNN | 10 | ✓ (2000 RoIs) | 2.9 | 0.42 | *N/A* |
| **R-FCN** | 0 | ✓ (2000 RoIs) | 0.46 | 0.17 | 79.3 |

Table 4: Comparisons on PASCAL VOC 2007 *test* set using **ResNet-101**. "Faster R-CNN +++" [10] uses iterative box regression, context, and multi-scale testing.

| | training data | mAP (%) | test time (sec/img) |
|---|---|---|---|
| Faster R-CNN [10] | 07+12 | 76.4 | 0.42 |
| Faster R-CNN +++ [10] | 07+12+COCO | **85.6** | 3.36 |
| **R-FCN** | 07+12 | 79.5 | 0.17 |
| **R-FCN** multi-sc train | 07+12 | 80.5 | 0.17 |
| **R-FCN** multi-sc train | 07+12+COCO | **83.6** | 0.17 |

Table 5: Comparisons on PASCAL VOC 2012 *test* set using **ResNet-101**. "07++12" [7] denotes the union set of 07 *trainval+test* and 12 *trainval*. [†]: http://host.robots.ox.ac.uk:8080/anonymous/44L5HI.html [‡]: http://host.robots.ox.ac.uk:8080/anonymous/MVCM2L.html

| | training data | mAP (%) | test time (sec/img) |
|---|---|---|---|
| Faster R-CNN [10] | 07++12 | 73.8 | 0.42 |
| Faster R-CNN +++ [10] | 07++12+COCO | **83.8** | 3.36 |
| **R-FCN** multi-sc train | 07++12 | 77.6[†] | 0.17 |
| **R-FCN** multi-sc train | 07++12+COCO | **82.0**[‡] | 0.17 |

**Comparisons with Faster R-CNN Using ResNet-101**

Next we compare with *standard* "Faster R-CNN + ResNet-101" [10] which is the strongest competitor and the top-performer on the PASCAL VOC, MS COCO, and ImageNet benchmarks. We use $k \times k = 7 \times 7$ in the following. Table 3 shows the comparisons. Faster R-CNN evaluates a 10-layer subnetwork for each region to achieve good accuracy, but R-FCN has negligible per-region cost. With 300 RoIs at test time, Faster R-CNN takes 0.42s per image, $2.5\times$ slower than our R-FCN that takes 0.17s per image (on a K40 GPU; this number is 0.11s on a Titan X GPU). R-FCN also trains faster than Faster R-CNN. Moreover, hard example mining [23] adds no cost to R-FCN training (Table 3). It is feasible to train R-FCN when mining from 2000 RoIs, in which case Faster R-CNN is $6\times$ slower (2.9s *vs.* 0.46s). But experiments show that mining from a larger set of candidates (*e.g.*, 2000) has no benefit (Table 3). So we use 300 RoIs for both training and inference in other parts of this paper.

Table 4 shows more comparisons. Following the multi-scale training in [9], we resize the image in each training iteration such that the scale is randomly sampled from {400,500,600,700,800} pixels. We still test a single scale of 600 pixels, so add no test-time cost. The mAP is 80.5%. In addition, we train our model on the MS COCO [14] *trainval* set and then fine-tune it on the PASCAL VOC set. R-FCN achieves 83.6% mAP (Table 4), close to the "Faster R-CNN +++" system in [10] that uses ResNet-101 as well. We note that our competitive result is obtained at a test speed of 0.17 seconds per image, $20\times$ faster than Faster R-CNN +++ that takes 3.36 seconds as it further incorporates iterative box regression, context, and multi-scale testing [10]. These comparisons are also observed on the PASCAL VOC 2012 test set (Table 5).

**On the Impact of Depth**

The following table shows the R-FCN results using ResNets of different depth [10], as well as the VGG-16 model [24]. For VGG-16 model, the fc layers (fc6, fc7) are turned into sliding convolutional layers, and a $1 \times 1$ convolutional layer is applied on top to generate the position-sensitive score

maps. R-FCN with VGG-16 achieves slightly lower than that of ResNet-50. Our detection accuracy increases when the depth is increased from 50 to 101 in ResNet, but gets saturated with a depth of 152.

| | training data | test data | VGG-16 | ResNet-50 | ResNet-101 | ResNet-152 |
|---|---|---|---|---|---|---|
| R-FCN | 07+12 | 07 | 75.6 | 77.0 | 79.5 | 79.6 |
| R-FCN multi-sc train | 07+12 | 07 | 76.5 | 78.7 | 80.5 | 80.4 |

**On the Impact of Region Proposals**

R-FCN can be easily applied with other region proposal methods, such as Selective Search (SS) [28] and Edge Boxes (EB) [29]. The following table shows the results (using ResNet-101) with different proposals. R-FCN performs competitively using SS or EB, showing the generality of our method.

| | training data | test data | RPN [19] | SS [28] | EB [29] |
|---|---|---|---|---|---|
| R-FCN | 07+12 | 07 | **79.5** | 77.2 | 77.8 |

## 4.2 Experiments on MS COCO

Next we evaluate on the MS COCO dataset [14] that has 80 object categories. Our experiments involve the 80k *train* set, 40k *val* set, and 20k *test-dev* set. We set the learning rate as 0.001 for 90k iterations and 0.0001 for next 30k iterations, with an effective mini-batch size of 8. We extend the alternating training [19] from 4-step to 5-step (*i.e.*, stopping after one more RPN training step), which slightly improves accuracy on this dataset when the features are shared; we also report that 2-step training is sufficient to achieve comparably good accuracy but the features are not shared.

The results are in Table 6. Our single-scale trained R-FCN baseline has a *val* result of 48.9%/27.6%. This is comparable to the Faster R-CNN baseline (48.4%/27.2%), but ours is 2.5× faster testing. It is noteworthy that our method performs better on objects of *small* sizes (defined by [14]). Our multi-scale trained (yet single-scale tested) R-FCN has a result of 49.1%/27.8% on the *val* set and 51.5%/29.2% on the *test-dev* set. Considering COCO's wide range of object scales, we further evaluate a multi-scale *testing* variant following [10], and use testing scales of {200,400,600,800,1000}. The mAP is 53.2%/31.5%. This result is close to the 1st-place result (Faster R-CNN +++ with ResNet-101, 55.7%/34.9%) in the MS COCO 2015 competition. Nevertheless, our method is simpler and adds no bells and whistles such as context or iterative box regression that were used by [10], and is faster for both training and testing.

Table 6: Comparisons on MS COCO dataset using **ResNet-101**. The COCO-style AP is evaluated @ IoU $\in [0.5, 0.95]$. AP@0.5 is the PASCAL-style AP evaluated @ IoU = 0.5.

| | training data | test data | AP@0.5 | AP | AP small | AP medium | AP large | test time (sec/img) |
|---|---|---|---|---|---|---|---|---|
| Faster R-CNN [10] | train | val | 48.4 | 27.2 | 6.6 | 28.6 | 45.0 | 0.42 |
| **R-FCN** | train | val | 48.9 | 27.6 | 8.9 | 30.5 | 42.0 | 0.17 |
| **R-FCN** multi-sc train | train | val | 49.1 | 27.8 | 8.8 | 30.8 | 42.2 | 0.17 |
| Faster R-CNN +++ [10] | trainval | test-dev | **55.7** | **34.9** | 15.6 | 38.7 | 50.9 | 3.36 |
| **R-FCN** | trainval | test-dev | 51.5 | 29.2 | 10.3 | 32.4 | 43.3 | 0.17 |
| **R-FCN** multi-sc train | trainval | test-dev | 51.9 | 29.9 | 10.8 | 32.8 | 45.0 | 0.17 |
| **R-FCN** multi-sc train, test | trainval | test-dev | **53.2** | **31.5** | 14.3 | 35.5 | 44.2 | 1.00 |

## 5  Conclusion and Future Work

We presented Region-based Fully Convolutional Networks, a simple but accurate and efficient framework for object detection. Our system naturally adopts the state-of-the-art image classification backbones, such as ResNets, that are by design fully convolutional. Our method achieves accuracy competitive with the Faster R-CNN counterpart, but is much faster during both training and inference.

We intentionally keep the R-FCN system presented in the paper simple. There have been a series of orthogonal extensions of FCNs that were developed for semantic segmentation (*e.g.*, see [2]), as well as extensions of region-based methods for object detection (*e.g.*, see [10, 1, 23]). We expect our system will easily enjoy the benefits of the progress in the field.

## Footnotes

[3]Although joint training [19] is applicable, it is not straightforward to perform example mining jointly.

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
