[Reviews · NeurIPS 2016]

Reviewer 1

Summary

The paper proposes a major simplification to the currently dominating approach of object detection: rather than using a region proposal mechanism to trigger the scoring of candidate object hypotheses through additional fully convolutional layers, the authors propose instead to push the 'fully convolutional' idea all the way, and use a region proposal mechanism only at the end of the detection pipeline - which is trained end-to-end so as to provide properly adapted inputs to the region scoring module. This is a quite important simplification that also accelerates and improves performance - it is the kind of simple ideas one wishes one had..

Qualitative Assessment

The paper is very well written - and will certainly become the next standard thing in detection. I am just not too sure whether the technical aspect is novel enough to support an oral at nips (the top-layer scoring function is not too different from the spatial pyramid pooling variant of Lampert et al in CVPR 2008), but this is certainly a paper that should be accepted. Some questions that the authors could address in the final version of the paper: - It would be useful to report results with VGG, at least for VOC - so that one can have a feel of what the ResNet101 gives as improvement. This is clearly orthogonal to the contribution of the paper, but it would be useful to have this number (since people will want to compare apples with apples when referring to your work, and may want to stick to their vgg-based systems). - Some experiments with an end-to-end trainable region proposal network would be useful; it is reported that this is 'hard to combine with OHEM' - but how about just using it without the hard example mining? - Does placing the RPN at an earlier stage help or decrease performance? In the RCNN system I think it was the case that using conv5 features for bounding box regression was better than using the fc7 features (thanks to better localized features apparently). Do you have any experience with that?

Confidence in this Review

2-Confident (read it all; understood it all reasonably well)


Reviewer 2

Summary

Currently best-performing object detection approaches operate by first generating a set of candidate bounding boxes, and then independently processing each candidate. The second stage of processing operates on the feature representation obtained by performing a pooling operation on the feature maps. The computational cost of the second stage grows linearly with the number of candidate bounding boxes, and incurres substantial cost since it typically relies on a fully connected neural network to generate the final region score. The paper proposes an approach to substantially reduce the cost of the second processing stage without loss in performance. To that end it is proposed to introduce a set of intermediate prediction targets that coarsely identify position of each image location with respect to object bounding box of a particular class (i.e. there are 7x7=49 such intermediate labels used in the paper). These intermediate "relative-location" labels can be automatically derived from ground-truth bounding boxes. During prediction confidence scores of these labels (denoted as "position sensitive score maps") are used to derive confidence scores for bounding box proposals with minimal computation cost (essentially summation over corresponding bounding box regions). This allows for efficient computation of bounding box proposal scores since the computation of position-sensitive score maps is done only once.

Qualitative Assessment

Object localization is one of the most competitive areas in computer vision. This paper builds on the best performing Faster-RCNN approach and demonstrates how to further speed it up by sharing most of the computations between proposal. I think this is a modification that might have substantial impact. The reason I am giving "poster" ranks is that I feel that the paper is to a large extent incremental over existing work. * I would like to ask the authors to comment on the case when image includes multiple overlapping objects of the same class. It seems that approach does not generalize well to this case because it requires each locations to have unique positional label within the object bounding box and this requirement is not satisfied for multiple overlapping boxes. * The approach allows to push computation to the stage prior to evaluating proposals. This means that it becomes practical to consider larger number of proposal. Is it beneficial to evaluate larger number of bounding box proposals? What is the recall for the case of 300 proposals per image as considered in the paper? * It appears that the spatial accuracy of the proposed method is related to the resolution of the position-sensitive score maps. Based on Fig. 3 and Fig. 4 these seem to be rather high-resolution. Does the proposed architecture include any form of up-convolution (i.e. deconvolution / up-sampling) layers to achieve this? Alternatively, what is the effect of resolution of the position-sensitive score maps on the localization accuracy?

Confidence in this Review

2-Confident (read it all; understood it all reasonably well)


Reviewer 3

Summary

This paper introduces a method for performing object detection via region-based fully convolutional networks. The main contribution is to define an architecture which offers a good trade-off between detection efficiency and accuracy.

Qualitative Assessment

The proposed approach is dedicated to improve the efficiency of state-of-the-art object detection approaches [8,6,18], which use two different networks (one for region proposal, one for region classification). The authors introduce an architecture based on a single fully convolutional network dedicated to share feature computation at every layer, which enables an important speed-up while maintaining near state-of-the-art detection performances. This is achieved by incorporating “position-sensitive score maps and Position-sensitive RoI pooling” on top of fully convolutionnal networks used for image classification (e.g. ResNet). On the positive side: - The experimental section is strong. Using the same baseline network ResNet-101 (ILSVRC’15 winner), several baseline methods are compared to the proposed method, as well as state-of-the-art methods for object detection, e.g. Faster R-CNN+++ [9]. - The authors show the significant boost of the proposed method compared to baseline (naïve) fully convolutional methods, and report similar performances compared to Faster R-CNN+++ [9], while being from 2,5 to 20 times faster. - Experiments are systematically conducted in challenging and modern datasets (ImageNet, VOC, MS COCO) On the negative side: - The paper is essentially experimental, since the main contribution relies to the new fully convolutional architecture design and the introduction of “position-sensitive score maps and Position-sensitive RoI pooling” to achieve this goal. -- A more formal justification of the proposed architecture would be a plus. For example, it would be interesting to give elements on why the position sensitive score maps have such an empirical edge over standard RoI pooling methods performed on top on convolutions layers (i.e.the naïve Faster RCNN variant evaluated in the experiments). - More generally, the presentation of the paper could be improved. -- The introduction starts with very technical issues, which are not necessarily appropriate for the general NIPS audience. -- No clear contribution of the paper is given (although the main novelty relies in terms of model architecture) -- The explanation of the Position-sensitive score maps and Position-sensitive RoI pooling is not fully satisfactory. In particular, Figure 1 and Figure 2 are redundant and could be improved to better explain how region features are pooled in a position-wise and class-wise manner. --- Figure 4, although interesting, would better fit the introduction section, but needs a more in depth and simpler explanation of the approach rationale. As a conclusion, the paper introduces a new method for object detection with strong experimental evidence, by featuring a good trade-off between efficiency and accuracy. However, the method could benefit from further justifications in terms of architectural choices, and from more pedagogical introduction of the author’s ideas.

Confidence in this Review

2-Confident (read it all; understood it all reasonably well)


Reviewer 4

Summary

This paper is an important extension of recent seminal object detection work Faster R-CNN proposed by Ren et al [18]. The main contribution of the paper is to redesign Faster R-CNN into a fully convolutional variant R-FCN through introducing so called position-sensitive score maps. Benefiting from the property of fully convolutional architecture, the proposed method can not only bring about 2.7 & 2.5X speeding up in training & testing, but also can obtain on par performance with standard Faster R-CNN method. Extensive experimental results are also provided to show its effectiveness and to analyze various choices regarding the network design, training and testing.

Qualitative Assessment

Faster R-CNN [18] is one of the latest top-performing object detection methods on several popular benchmark datasets such as PASCAL VOC, Microsoft COCO and ImageNet. The main novelty of this paper is to redesign Faster R-CNN method into a fully convolutional variant R-FCN through introducing so called position-sensitive score maps. Note that the basic idea of position-sensitive score maps is extended from instance-sensitive score maps proposed by Dai et al. [3] recently. However by properly combining it with Faster R-CNN [18] and powerful ResNet [9], the proposed R-FCN method can not only bring about 2.7X & 2.5X speeding up in training & testing, but also can obtain on par performance with standard Faster R-CNN method. Extensive experiments are conducted for evaluation and analysis. Overall, this paper is a good submission. But it can be further improved if the following concerns can be addressed. - Obviously, the performance improvement of R-FCN is not just from position-sensitive score maps. The modification of ResNet with “hole algorithm” [2, 15] and the improved training with OHEM [22] also play important role. In Table 4, 5, 6 and two additional Tables (without captions) on page 8, it is important to clearly clarify that OHEM is used or not used in R-FCN. According to the results shown in Table 3, I think OHEM should be used for all results of R-FCN in above mentioned Tables. This is also important for fair comparisons with standard Faster R-CNN and Faster R-CNN +++ since they do not use OHEM. - As the authors describe that Faster R-CNN +++ further incorporates iterative BBR, global context and multi-scale testing, yet R-FCN incorporates “hole algorithm” and OHEM (and also multi-scale testing on Microsoft COCO dataset), how about object detection accuracy of R-FCN when using the completely same settings regarding its best results or all tricks, e.g., adding iterative BBR and global context. This is important to show the best potential of R-FCN. Will this provide best results on PASCAL VOC and Microsoft COCO datasets? - In line 250 and 251, the authors claim that R-FCN performs better on the objects of small sizes (in Microsoft COCO dataset) than standard Faster R-CNN. But according to Table 6, two methods show very similar performance, i.e., ~0.5% accuracy difference with respect to mAPs, does this mean that R-FCN performs less better/similar on the objects of middle sizes or large sizes? If yes, this is somewhat confusing because larger object instances usually activate more in feature maps compared with smaller object instances. - Note that R-FCN still adopts 4-step (even 5-steps alternating training strategy on Microsoft COCO dataset) proposed in standard Faster R-CNN, it would be interesting if the authors can provide a single-run training scheme and make it more straightforward to train the detector, as like YOLO [17] and SSD [14]. - A minor typo should be corrected. According to Table 6, in line 252, “51.5%/29.2%”->”51.9%/29.9%”. - One related work (Gidaris and Komodakis, Object detection via a multi-region & semantic segmentation-aware cnn model, in ICCV 2015), in which the use of multi-region contextual information was first explored, should be cited. - The authors provide detailed info on training and testing, it will be better if the entire training time can be given for each benchmark dataset.

Confidence in this Review

3-Expert (read the paper in detail, know the area, quite certain of my opinion)


Reviewer 5

Summary

In this paper, a deep object detection network named Region-based Fully Convolutional Network is proposed. In this network, position-sensitive score maps are incorporated as the FCN output to cross the gap between classification tasks and object detection. And, to extract information from such score maps, selective RoI pooling strategy is proposed. Overall, this is an impressive work with unique innovation. The experiments conducted on PASCAL VOC and COCO demonstrate its good performance.

Qualitative Assessment

Paper Strengths: 1. Both the training and testing procedures are faster than Faster R-CNN. Average 0.45 second per image for training and 0.17 second per image for testing, which is about 2.5-20x faster than faster R-CNN. 2. Position-sensitive score maps and selective pooling links the classification network and the object detection task. 3. Multiple popular network can are applicable in the R-FCN framework. And the end to end training makes the proposed method easy to follow. 4. The experiment is pretty exhaustive and comprehensive. It shows the effectiveness, efficiency and compatibility of the proposed method. Paper Weakness: 1. The organization of this paper could be further improved, such as give more background knowledge of the proposed method and bring the description of the relate literatures forward. 2. It will be good to see some failure cases and related discussion.

Confidence in this Review

2-Confident (read it all; understood it all reasonably well)


Reviewer 6

Summary

The authors introduce an object detection method call R-FCN that shows similar performance to prior methods like Faster R-CNN, but is faster to train and to evaluate. This is achieved bu using a fully convolutional pipeline and a novel combination of position-sensitive score map and position sensitive RoI layer.

Qualitative Assessment

I enjoyed the paper a lot and I am happy to see the benefits of a Fully-Convolutional approach to be applied to object-detection. The speedup you achieve is impressive. In section 4.1 On the Impact of Region Proposals you state that "R-FCN performs competitively using SS or EB", but the Table does not include any other system using SS or EB to compare against.

Confidence in this Review

2-Confident (read it all; understood it all reasonably well)